# Orchard Management and Incorporation of Biochemical and Molecular Strategies for Improving Drought Tolerance in Fruit Tree Crops

**DOI:** 10.3390/plants12040773

**Published:** 2023-02-08

**Authors:** Sama Rahimi Devin, Ángela S. Prudencio, Sayyed Mohammad Ehsan Mahdavi, Manuel Rubio, Pedro J. Martínez-García, Pedro Martínez-Gómez

**Affiliations:** 1Department of Horticultural Science, College of Agriculture, Shiraz University, Shiraz 7144165186, Iran; 2Department of Plant Breeding, CEBAS-CSIC, P.O. Box 164, Espinardo, 30100 Murcia, Spain

**Keywords:** drought, fruit yield, climate change, gene overexpression, CRISPR

## Abstract

Water scarcity is one of the greatest concerns for agronomy worldwide. In recent years, many water resources have been depleted due to multiple factors, especially mismanagement. Water resource shortages lead to cropland expansion, which likely influences climate change and affects global agriculture, especially horticultural crops. Fruit yield is the final aim in commercial orchards; however, drought can slow tree growth and/or decrease fruit yield and quality. It is therefore necessary to find approaches to solve this problem. The main objective of this review is to discuss the most recent horticultural, biochemical, and molecular strategies adopted to improve the response of temperate fruit crops to water stress. We also address the viability of cultivating fruit trees in dry areas and provide precise protection methods for planting fruit trees in arid lands. We review the main factors involved in planting fruit trees in dry areas, including plant material selection, regulated deficit irrigation (DI) strategies, rainwater harvesting (RWH), and anti-water stress materials. We also provide a detailed analysis of the molecular strategies developed to combat drought, such as Clustered Regularly Interspaced Short Palindromic Repeat (CRISPR) through gene overexpression or gene silencing. Finally, we look at the molecular mechanisms associated with the contribution of the microbiome to improving plant responses to drought.

## 1. Introduction

Global warming, a decrease in precipitation, and population growth (the United Nations predicts that by 2050 the population could exceed 10 billion)—resulting in a 60% increase in the demand for food—are set to pose increasingly serious challenges to world food production, and especially temperate fruit production, in the coming years. Drought is one of the most destructive abiotic stresses for crop productivity, affecting more than a third of the world’s population. The water crisis afflicts human societies, affecting the economy and both human and animal health. In recent decades, the expansion of drylands has become a tremendous issue for many countries [1]. It is estimated that drylands represent over 40% of terrestrial land surfaces, which can be detrimental to the economy and society of approximately 600 million people in all corners of the world, especially in South & East Asia, North Africa, and the Middle East [2]. The ratio of annual precipitation (P) to annual Potential Evapotranspiration (PET) in drylands is less than 0.65 [1]. Drylands are classified into hyper-arid, arid, semi-arid, and dry sub-humid lands [3].

Due to the fact that the average temperature is increasing worldwide, it is predicted that changes in precipitation patterns will occur in the following century, intensifying drought in many regions. This climate change affects water availability in orchards [4].

The Food and Agriculture Organization (FAO) estimates that drought resulted in a direct loss of nearly USD 30 billion for agriculture in developing countries from 2005 to 2015. Drought stress, as the major abiotic stress in many regions, limits the fruitfulness of horticultural crops and agricultural development in arid and semi-arid regions [5]. Many orchard irrigation systems are not optimized for severe abiotic conditions, so the water requirements for plant growth are not supplied at the critical stages of growth [6]. The ultimate aim of commercial goals is fruit yield; to produce a successful yield, basic conditions must be met [7]. Water shortage is more problematic in the final period of fruit growth and can result in fruits that are smaller than usual and not suitable for the market [8].

There is great potential for developing the horticultural sector in arid lands worldwide [9]. Improved technologies, such as regulated Deficit Irrigation (DI) [10], can boost low productivity in arid areas [9]. It is essential to find strategies to improve the resistance and tolerance of fruit species to drought by identifying their physiological, biochemical, and molecular parameters. Using drought-resistant species as rootstocks is one of the most optimal solutions to plant production in arid regions [11]. The tree canopy, for instance, plays an important role in tree adaptation to drought conditions, so dwarf cultivars with dense canopies are more resistant to water stress than broad-canopied cultivars [12]. Mulching can enhance the percentage of soil moisture and reduce water evaporation from the soil [13]. Under drought conditions, heavy summer pruning can reduce the negative effects of water stress on fruit growth by decreasing canopy transpiration [14]. Another method that can increase fruit growth in dry conditions is to thin fruits heavily [14]. Another gardening technique is net shading, which can help decrease water needs [8]. Another interesting water-saving technology called “cocoon” has proven to boost seedling survival in all parts of the world in fruit tree crops such as mango [15]. Cocoon is small water reservoir technology which is used for plant growth in the dry season and thereby eliminates the need for irrigation. Plant tolerance to abiotic stresses may also be enhanced by the exogenous use of desirable molecules such as humic acid, amino acids, proline, etc. [16]. Finally, semi-open mechanical structures in a half-circle shape, called half-moons, gather and keep runoff water, thereby ameliorating soil penetration. It is a semi-open mechanical structure with the shape of a half-circle that collects and holds runoff water and improves soil infiltration. Thus, it contributes to recovering and restoring the fertility of the encrusted soil for agronomic and agroforestry purposes. These structures have been largely assayed in native fruit trees from Burkina Faso in Africa [17].

As global human population growth has increased, cropland expansion has risen as well; this expansion imposes intense pressure on water resources [7]. In order to sustain the growing world population, efficient water management strategies and systems are vital [18]. To ameliorate the adverse effects of abiotic stresses on crops and ensure the livelihood of the population, new sustainable technologies from water management systems to genetic engineering must be improved.

The main objective of this review Is to integrate and discuss the most recent horticultural strategies adopted to improve the response of temperate fruit crops to water stress, addressing the viability of cultivating fruit trees in dry areas, and to provide precise protection methods for planting fruit trees in these areas. Emphasis will be placed on the analysis of the physiological, biochemical, and molecular aspects associated with drought tolerance. These new approaches (Figure 1) are of particular importance because of the possibility of growing fruit trees in arid areas and improving optimal yield and fruit quality.

## 2. Orchard Establishment and Horticultural Management

In establishing new orchards, the cultivation needs for the specific fruit tree must be considered. The tree should be compatible with the soil, water, and weather conditions [19,20].

### 2.1. Rootstock and Variety Selection

Selecting drought-tolerant species is a promising solution in arid and semi-arid regions [21]. These kinds of cultivars are more efficient because they are able to withstand DI with minimal impact on fruit quality and yield and thus contribute to water saving [22]. When a tree maintains both its water transport capacity and carbon supply under drought stress, high performance of the physiological mechanisms occurs [23]. Wild relatives of fruit tree species are candidates for breeding future crops and have attracted the attention of scientists researching drought [11]. Many native wild tree species have been found to perform better under drought conditions than cultivated species [24]. In these species, a lower vulnerability to xylem embolism leads to a more robust seasonal pattern of photosynthesis [11]. Wild-relative species are recommended for developing straightforward rootstocks or creating interspecific hybrids, with the aim of increasing fruit tree resistance to drought stress [24].

To increase water-use efficiency (WUE), Jiménez et al. [25] indicated that the rootstocks ‘Cadaman’, ‘GF-677’, ‘Rootpac 20’ (*Prunus besseyi* Χ *P. cerasifera*), and ‘Rootpac R’ (*P. cerasifera* Χ *P. dulcis*) grafted with ‘Catherina’ peach increased the level of proline (in roots and leaves), sorbitol (in leaves), and raffinose (in roots) under water-deficit conditions [25]. Interspecific hybrids of Prunus, on the other hand, had a decrease in water potential, photosynthesis, and transpiration under drought conditions, and a significant increase in enzyme activity in recovery periods, when plants were well-watered [26]. In another study conducted by Gelly et al. [27], peach grafted onto ‘GF-677’ performed better under controlled water deficit in phase II of fruit growth, with increased soluble solids and improved fruit quality. This study shows that trees can save water in this stage of fruit growth without decreasing fruit size [26].

Using wild varieties as rootstocks is also recommended for pome fruit trees [7]. As a case in point, Al Maarri et al. [28] showed that the Syrian pear (*Pyrus syriaca*), the only wild species in Lebanon and Turkey, is drought tolerant and has higher grafting compatibility with *Pyrus communis* than quince (*Cydonia oblonga*) rootstocks [7].

### 2.2. Canopy Architecture and Pruning

Pomologists have proved that the canopy architecture can play a key role in the adaptation of trees to abiotic conditions [12]. When it comes to drought tolerance, dwarf cultivars with compact crowns are more tolerant to drought than the wide-crowned varieties and delay dehydration [29]. In such a tree canopy, the leaf size is smaller, leading to a decrease in transpiration and improved energy saving [12]. High density crowns also protect the inner leaves from solar radiation and improve the microclimate in the tree crown [12].

The adverse impacts of water deficit on fruit yield can be alleviated by changing the canopy architecture and removing the outer 30% of all major branches via pruning; severe summer pruning (SSP) is especially effective [8,30]. When the canopy is pruned, the leaf area is decreased, which reduces the water demand of the tree and ultimately enhances the tree water status [8]. In fact, pruning reduces the hydraulic conductivity and transpiration rate, helping trees delay drought symptoms [31]. Selecting an appropriate pruning method for each tree can therefore be an important water-saving strategy during drought seasons [32].

### 2.3. Flower and Fruit Thinning

Another technique for ameliorating the damaging effect of drought stress is removing flowers or young fruits, known as thinning [14]. Fruit thinning reduces competition between fruits and increases the amount of carbohydrates for the remaining fruits and the tree, significantly improving the tree water status under water stress conditions due to the additional root growth and stomatal closure [33]. As a result, the growth rate of the remaining fruits is increased [8]. Although heavy fruit thinning reduces the total productivity rate of the tree, it increases commercial fruit size classes [8]. In addition, thinning improves fruit quality [34]. It should also be mentioned here that the severity of thinning ought to be regulated according to the intensity of the water restrictions [35]. That is why thinning at the time of water shortage does not have high efficacy in terms of improving fruit quality [8].

Recent studies have focused on investigating the interaction between different materials and methods in plants in order to introduce more efficient and practical strategies to withstand DI [30]. Chen et al. [36], for instance, suggested that using mulch together with pruning may have a synergistic and superior effect in terms of water conservation. In other research conducted by Lopez et al. [14], the combination of severe summer pruning and heavy fruit thinning in peach trees at the onset of stage III (the final stage of fruit growth) was found to improve the tree water status. In hot regions such as southern Morocco, almond trees are grown in oasis-like climate conditions. Under these conditions, almond trees are planted as a middle layer of vegetation that forms an oasis under the palm trees [37]. In this agroforestry system, many trees from different species are densely planted in a specific intercropping system to protect the plants growing under a layer of date palms from the high temperatures and warm winds characteristic of dry climates and to increase soil nutrients [37]. These tree-based intercropping systems have been also assayed in more humid regions in Eastern Canada [38].

### 2.4. Net Shading, Mulching and Biochar Application

Net shading is another horticultural technique that can help decrease the water demand in fruit trees [8]. By installing shades over the trees, the photosynthetic photon flux is reduced [39]. This decreases the total temperature of the canopy and reduces water use [40]. Extensive research has been conducted to prove the effects of net shading on crops [41,42,43]. Nicolas et al. [44], for instance, showed that transpiration in shaded apricot ‘Bulida’ trees was lower than in trees exposed to direct solar radiation. After rewatering, the shaded trees recovered faster than the exposed ones [8]. It is important to choose a shading net with an appropriate color [39]. For instance, the shading potential of a white net is 25%, and that of a black one is 40%. According to a study conducted by Villalobos-Soublett et al. [39], although both types of shading nets decreased the damaging effects of solar radiation in vines, a higher bunch weight and number of berries per bunch were obtained under white shading, while a higher pruning weight was observed in the black shading treatments.

The use of mulches is another practical method for decreasing the evaporation of water from soil surfaces in order to alleviate soil desiccation in arid areas [45]. Crop residues (e.g., grass and straw), plastic film (e.g., black polyethylene), gravel-sand, rock fragments, volcanic ash, poultry and livestock litter, paper pellets, city rubbish, and other materials can be used as mulch [13].

Some mulches—such as straw mulching—are thought to be relatively more environmentally friendly than others, such as the widespread use of plastic film [30]. Mulch efficiency depends not only on its abilities to preserve water, but also on its potential to be applied in large areas, restrain weed growth, and regulate soil temperature; it should also have a long lifespan and excellent diathermancy and air permeability [46]. Ultimately, mulch can improve fruit tree growth by improving net photosynthesis rate, stomatal conductance, and intercellular CO_2_ concentration [30].

Biochar is a mainly steady resistant organic carbon compound obtained by heating biomass at temperatures typically between 300 and 1000 °C in an oxygen-limited environment [47]. Biochar can increase soil water conservation [48] and fertility [49] and is therefore of great value for soil management in orchards located in arid areas. The capacity of biochar-enriched soil to retain water was 18% higher than that of biochar-free soil [50]. The use of biochar in potted *Pyrus ussuriensis* seedlings showed significantly delayed soil moisture loss and effectively changed the chlorophyll fluorescence parameters, increasing the photochemical efficiency of the photosystem II [51].

### 2.5. Deficit Irrigation

Deficit Irrigation (DI) is a water-saving irrigation strategy used to manage limited water resources for fruit trees in dryland areas [22]. Improved innovative and precise DI can mitigate and even eliminate the damaging effects of drought on crop yield and quality [52]. For instance, Santos et al. [53] reported that DI enhanced fruit quality and flavor in grapes. Other research has shown that the concentration of total soluble solids and acids increases in several fruit species by using DI [54]. This does not only save irrigation water but also leads to a decrease in fertilizer and pesticide application and thus prevents groundwater contamination [18]. Generally, woody plant vegetative growth is very sensitive to DI. Fruit growth (both cell division and cell size) is reduced when the branch length growth decreases in response to DI, but fruit firmness may increase, and this characteristic is important in pome and stone fruits [18].

Sustained deficit irrigation (SDI) originated in the 1970s with a strategy called deficit high-frequency irrigation (DHFI), which was very similar to SDI [55]. It was not successful when the soil water was not optimal [22]. SDI distributes the water deficit uniformly over the fruit season at any crop stage and is a practical strategy for preserving fruit quality under drought stress conditions based on setting aside the water deficit uniformly over the fruit season at any crop stage [52]. This is a type of DI strategy in which the irrigation water used at any given time during the season is less than the evapotranspiration (ETc) demands [52]. Applying less water than the evapotranspiration demands of the crop does not fill the roots and creates progressive stress in the plant during the season [56]. Monitored DI has been found to enhance fruit quality and increase the sugar content and minerals in mango fruit [56].

Regulated deficit irrigation (RDI) is a strategy in which irrigation is fully applied during critical periods for fruit trees—the drought-sensitive phenological stages—and limited during the non-critical periods—the drought-tolerant phenological stages [57]. This works because the sensitivity of fruit trees to water deficit is greater in some growing stages, and not necessarily during the whole growing season [58]. A portion of water can be supplied by rainfall during the non-critical periods, for instance [22]. The harmful effects of water shortage on fruit yield might be minimized by increasing irrigation water savings during particular periods [59]. This can also improve harvest quality [59]. Several studies have proved the efficacy of RDI. Lipan et al. [60], for example, reported that applying moderate RDI can increase the quality of almonds, producing a redder color, a higher fat and potassium content, and a greater unsaturated fatty acid concentration. According to the different water requirements of different phenological periods in fruit trees, the basic question in the RDI strategies was to carry out water deficit irrigation, saving it to a certain extent without affecting the basic growth of trees. The basic principle of this method was that the water demand of trees and the effects of water deficit on trees at different growth stages were different. In the non-critical stages, the amount of irrigation water of trees was less than that required.

The partial root-zone drying technique (PRD) involves irrigating only one part of the root zone, leaving the other side dry, and rewetting by shifting irrigation to the dry side through alternate furrow irrigation [61]. PRD can be effective because the roots trigger the synthesis of abscisic acid (ABA), leading to stomatal closure [62]. In addition, the lower cytokine levels and the higher xylem pH, as physiological responses to PRD, can also favor the closure of the stomata [63,64]. In grapevine, PRD has been shown to enhance the growth of grapevine roots [65]. For high crop yields and productivity in dryland areas, it is also crucial to know the periods in which specific fruit trees can best handle water stress [22]. DI inflicts a period of water stress whose severity and length are monitored [66]. This period is usually linked to the slower stages of fruit growth in which the tree is comparatively more resistant to water shortage [67]. This happens in stages II and III of fruit growth in almond trees, but in stage II for peach and plum trees [68]. In peach trees, RDI can decrease yields if the tree water status recovery is postponed after DI, especially when the water stress extends into stage III of fruit development [69]. In early-maturing peach trees, with a very short period from fruit set to harvest and a very long phenological period after harvest, DI should only be used in the post-harvest period so as not to affect yield and fruit quality [69]. Citrus and pome fruits have a critical period of rapid fruit growth, corresponding to a process of cell expansion, although some cell division can also occur in the beginning [52]. During this period, they are more sensitive to water deficit, which can limit fruit growth [58].

### 2.6. Rainwater Harvesting and Saving Technologies

Different techniques are used to harvest and save rainwater to protect the soil [70]. Such techniques in drier areas include Fanya juu terraces, Zai planting pits, Negarims, half-moons, and others [17,70]. Fanya juu are bench terraces with a sloping back that are built by digging and throwing soil upwards to make an embankment along the contour [70]. The origin is not clear, although such terraces were implemented on a large scale during colonial rule in Kenya [71]. Fanya juu terraces are suitable on slopes with a yearly rainfall of 500–1000 mm [72]. According to Wakindiki et al. [73], terraces decrease the slope and velocity of the ground flow and are constructed by digging a 60-cm wide trench along the contour. This has a significant impact on decreasing slope length, thus enhancing water infiltration and decreasing the erosion of soil and runoff from sloping farmland [70].

On the other hand, Zai pits are made by burying planting holes and filling them with organic matters during the dry season, to enhance soil moisture retention and available nutrients [17,74]. The diameter of the plant hole is 20 to 30 cm and the depth 10 to 15 cm [74], and there are usually between 12,000 and 15,000 per hectare [75,76]. After digging the holes, organic matter is added and then covered after the first rain [17]. The manure in the hole absorbs termites, and the termites dig holes in the ground, thus facilitating a deeper penetration of rainwater and runoff. Termites also transport nutrients from deeper layers to the ground surface [77]. Zai pits also increase greenery because they bring degraded land into agricultural production and may enhance planted tree density [78].

Negarims are tiny, diamond-shaped basins that are enclosed by embankments [79]. These basins are common in Kenya [71]. Negarims are used to establish fruit trees in drought regions with an annual precipitation of up to 150 mm, with a recommended slope of up to 5% [70]. They are made to resemble square embankments that have been rotated 45° from the contour to centralize surface runoff [80].

A half-moon is a semi-circular, mechanical structure that gathers and retains running water and improves soil permeability [17]. It thus helps to recover and restore the fertility of soil without coating it for agricultural purposes [76]. Half-moons are made using a 2-m compass beam with a diameter of 4 m and a height of 25 cm [17]. The distance recommended from the center of each half-moon is 8 m [81]. In addition, the growth of plants in the substrates ameliorates the crop yield on farms [82].

Finally, other materials used for generating water-retaining soil include grass strips, stone lines, trash lines, contour stone bunds, and Cocoons. Contour stone bunds are low lines of stones measuring 25 cm high and 35 cm wide on average [83]. These structures can enhance the tree cover density [78]. The Cocoon is a 100% biodegradable water-saving technology that increases seedling plantation survival rates (75–95%) in arid and semi-arid conditions [15]. This technology has been used successfully in many countries throughout the world [15]. Not only does it prevent evaporation, but it also increases plant growth and reduces irrigation requirements [15]. It is a cost-effective means to enhance land restoration [84]. The Cocoon has a water reservoir with a 25-L water-storing capacity that surrounds the young trees and feeds water to the soil at a slow and constant rate [85]. It is common to fill the reservoir at the planting stage [15]. This can provide water to the plant for around six months [85].

### 2.7. Plant Growth-Promoting Rhizobacteria (PGPR) and Arbuscular Mycorrhizal Fungi (AMF)

It has been reported that stress-adapted microorganisms from either the rhizosphere, internal tissues, or aerial parts of plants help plants cope with drought and promote growth through various mechanisms. These mechanisms include the accumulation of osmolytes, also lowering the level of ethylene inhibition by the enzyme aminocyclopropane-1-carboxylate deaminase (ACC) and providing unavailable nutrients for plants. *Azotobacter*, *Serratia*, *Bacillus*, *Pseudomonas*, and *Ochrobactrum* are microbial genera that are known as growth promoters under water stress [86].

Plant growth-promoting rhizobacteria (PGPR) are found around the plant root system and interact with plants, increasing their growth and thereby helping them maintain a promising water status in drought conditions [87]. Plant roots secrete organic compounds that are a source of carbon and attract PGPR [88]. PGPR increases the yield in some plants, including apple trees [89]. *Serratia plymuthica* and *Pseudomonas lini* rhizobacteria have been found to increase drought tolerance in jujube seedlings [90].

Arbuscular mycorrhizal fungi (AMF) establish symbiotic associations with the roots of approximately 80% land plant species. Such symbiotic associations help the host plants absorb soil water and nutrients in exchange for 20% of the host plant’s photosynthetic carbohydrates [91]. Studies have shown that the accumulation, synthesis, and degradation of proline in plants can be regulated by the AMF in response to salinity and drought stress [92]. In addition, AMF has been found to increase nutrient retention in the host plant and improve water relations under drought conditions in citrus plants [93]. Mycorrhiza (10 mL of spore suspension) mixed with soil also improved olive growth parameters [94].

## 3. Biochemical Treatments for Drought Tolerance and Their Physiological Effects

Plant resilience in response to environmental pressures may be increased by the exogenous application of favorable molecules [95]. Different chemicals have been used to activate plant mechanisms that improve survival and production after dehydration [96].

Ascorbic acid, for instance, is a well-known antioxidant used to regulate photosynthesis in plants [16,97]. The foliar use of ascorbic acid (250 ppm concentration) in two young peach tree cultivars (‘Scarletprince’ and ‘CaroTiger’) alleviated stress after short periods of water scarcity and rewatering [16]. Moreover, the treatment of roots with ascorbic acid improved root elongation under water stress in tall fescue [98]. The ascorbic acid applied by irrigation can increase productivity under drought conditions in the common bean [99].

Triazoles are chemical compounds belonging to the group of ergosterol biosynthesis inhibitors [100]. These chemicals, like hexaconazole, paclobutrazol (PBZ), triadimfon, and triazole, are applied as fungicides and also have different levels of plant growth-regulating effects [101]. They preserve plants faced with biotic and abiotic stresses [102] by increasing the proline content and antioxidant enzyme activities [100]. Former studies have reported decreased transpiration and enhanced drought resistance in some tree seedlings after applying triazoles [103]. Paclobutrazol (PBZ) is a triazole chemical that is widely applied in horticulture as a fungicide and plant growth regulator [104]. In ornamental plants, PBZ is used to decrease plant size, improve compaction, and enhance other functional aspects that help plants withstand biotic and abiotic stresses [104]. Furthermore, some authors have identified an anti-transpiration property of PBZ that clearly affects water relationships and the related biochemical and physiological changes [105]. Apple trees whose roots were soaked in PBZ (0.5 g), for instance, had a smaller decrease in the leaf water potential under water stress [106].

Chitosan, a marine cationic polysaccharide, has special bioactive attributes that make it an efficient scavenger of reactive oxygen species (ROS) [107]. The use of chitosan has therefore been recommended to decrease oxidative damage caused by water stress in plants [108], as observed in apple leaves [96]. The foliar treatment of apple seedlings with a chitosan solution (100 mg/L) before water stress was also found to reduce malondialdehyde production and electrolyte leakage in leaves, while enhancing the activities of the superoxide-dismutase (SOD) antioxidant enzyme [108].

Putrescine, spermidine, and spermine (Put, Spd, and Spm) are the most abundant polyamines (PAs)—organic polycations—and are involved in several growth processes in plants [109]. Polyamines can modify the size of the potassium channel and the size of pores in the plasma membrane of guard cells, thereby strongly regulating stomatal opening. In this way, PAs can control water loss in plants. Foliar application of Put, at an appropriate level, can trigger the biosynthesis of osmotic adjustment substances, such as proline, although there is also evidence of a detrimental effect in wheat under drought stress [110]. On the other hand, Spd deferred senescence in jack pine under water stress [111]. Among the three main endogenous PAs, Spm was most strongly related to drought resistance in apples and cherry tomatoes [110]. Moreover, treatment with Spm (1 mM) induced drought tolerance in citrus plants grown in vitro [112].

The DL-β-aminobutyric acid (BABA) compound has also been shown to protect plants against environmental stresses [113]. Grapevines stimulated by BABA accumulated callose and lignin and were protected against pathogen attack and water stress [96].

Glycine betaine (GB) is an osmoprotectant organic solute whose foliar application (50 mM) has a positive effect on papaya drought reaction, leading to a decrease in water stress [114]. GB can be involved in preventing the accumulation of ROS, protection of the membrane, conservation of the photosynthetic system, and transcriptional changes of stress-related gene activation [99,100]. In papaya, GB-treated plants retained enough water in drought conditions by controlling the movement of stomata and improving the osmotic adjustment by making compatible solutes available [114].

The plant hormone ABA (abscisic acid) is linked to the plant response to water stress [115,116]. When treated with ABA and BABA (1.0 mM each), the water status of apple trees improved, and there was also approximately 80% shoot growth following re-irrigation after water stress [96]. ABA can also be an effective antitranspirant when used on black spruce seedlings [117].

Kaolin is a kind of clay found in nature and is non-toxic when sprayed on plants, providing a thin coating that protects the plants against abiotic and biotic stress [18]. The covering produced by this material is porous, so it does not inhibit gas exchange or leaf stomatal closure and allows active photosynthetic radiation [118]. Kaolin spray has been shown to reduce leaf temperature by enhancing leaf reflectance and to decrease the rate of transpiration more than photosynthesis in some plant species growing under high levels of solar radiation [13]. In apple trees under DI, vegetative growth attributes such as stem length and diameter decreased, but kaolin improved these parameters [119]. Moreover, the use of kaolin enhanced the accumulation of anthocyanins and sugar in the fruit and even improved the apples’ red color [18]. Kaolin also enhanced the yield in pomegranate and walnut trees [120,121]. In grapevines, treatment with kaolin (5%) significantly reduced leaf lipid peroxidation under water stress [122]. The application of 6% kaolin in pomegranate trees under water stress also reduced the fruit cracking percentages [123].

Silicon is reported to enhance drought resistance in plants by retaining photosynthetic activity, plant water balance, and vascular structure under high transpiration [124]. The use of silicon has also been found to increase leaf water potential under drought conditions [13]. Silicon is also effective in protecting plants from destructive oxidative reactions, thereby increasing the capability of mango trees to resist abiotic stress in arid areas [125]. Spraying 5% silicon as an antitranspirant and treating banana plants with 60% dehydration is a promising way to decrease the total amount of the water needed for irrigation during the growing season. In addition, the growth parameters increased [126].

Brassinosteroid (BR) phytohormones can also provide tolerance to abiotic and biotic stresses [127]. Spraying apple trees with 0.05 ppm BRs before imposing stress can minimize the harmful effects of drought [128].

Antitranspirant compounds enhance drought resistance by reducing water transpiration in plants [129]. According to their roles, they can be categorized into two main groups: physical and physiological antitranspirants [130]. Physical antitranspirants containing latexes, polymers, or waxes cover the surface of leaves and thereby reduce plant water loss [129]. For instance, β-pinene polymer produced a remarkable decrease in water use (40%) in peach trees [131]. Physiological antiperspirants containing ABA or other chemicals reduce transpiration by stimulating plants to block stomata [132]. External use of ABA has increased drought tolerance in different horticultural crops [130,133].

Melatonin is an indole-based structure found in various organisms [134]. The application of melatonin leads to higher water retention in leaves, lower electrolyte leakage, and higher photosynthesis and starch collection under stress conditions [135,136]. Previous reports also suggested that treating apple trees with melatonin (100 µM) can retain a higher rate of carbon dioxide assimilation under drought conditions [137].

The application of superabsorbent polymers or hydrogels (synthetic polyacrylamide) can enhance the water holding capacity and decrease water deficiency by permeation [138]. A medium or high dose of hydrogel compound (1000 to 1500 g/tree) increases the yield and fruit quality in citrus trees, which may be due to the increase in nutrients and water accessibility [139]. The application of a hydrogel (0.4%) concentration enhanced plant survival in citrus rootstock seedlings that underwent several drying cycles [140].

Ethylene buildup increases the permeation of cell membranes under drought stress, and, in turn, the production of ethylene is prevented by aminovinylglycine (AVG) [141]. In line with this, the application of AVG in the root zone has been found to decrease transpiration under water stress [142].

Salicylic acid, a plant growth regulator, has been shown to regulate different metabolic and physiological activities in plants against environmental stresses, and it can be used in combination with AVG [143,144]. The use of a foliar SA (100 mg/L) treatment decreases drought stress in apricots [145].

Phosphatidylcholine is a type of lipid found in the cell membrane of many eukaryotes [146]. The use of phosphatidylcholine in the soil at a concentration of 500 mg/L can ameliorate the drought resistance of peach seedlings and decrease cell membrane damage due to water stress [6]. Phosphatidylcholine-sprayed peach trees were similar to well-irrigated trees, but drought-stressed trees that were not supplemented with phosphatidylcholine showed lower photosynthetic rates [6].

Nitric oxide (NO) is an unstable free radical with many biological functions in plants [147]. It can act as an antioxidant and scavenge ROS, thus preserving plants from abiotic stresses [148]. Apple seedling leaves were sprayed with the NO donor sodium nitroprusside (SNP), which reduced drought-induced ion leakages and the accumulation of soluble proteins [149].

Plant biostimulants consist of various organic and inorganic substances or microorganisms that can enhance fruit quality, nutrient uptake, and resistance to abiotic or biotic stresses [150]. The use of biostimulants, specifically seaweed extract, increased the weight of grape berries under water stress [151].

Seaweed and seaweed extracts have long been applied as fertilizers [152]. Seaweed extracts contain useful nutrients and can enhance plant growth, photosynthetic activity, and resistance to different stresses and thus improve fruit yield [151]. These substrates have different hormones and organic compounds that can help plants grow [153]. The seaweed *Ascophyllum nodosum*, used at 5 and 10 mL·L^−1^ as either a soil drench or foliar spray, had a significant effect on plant-water relations and may be an effective means for improving water stress resistance in citrus trees [152].

Amino acids are obtained by protein hydrolysis and have an important role in tree growth and conservation against abiotic stresses [151]. They are important to nitrogen metabolism and biosynthesis of chlorophyll [150]. The amino acid proline functions as an osmoprotectant as it binds to the hydrogen bonds of proteins, conferring structural stability and thus protecting proteins from denaturation under stress [154,155]. GABA (γ-aminobutyric acid) is a non-protein amino acid involved in different physiological processes, and it protects plants against environmental stresses like drought by enhancing leaf turgor and osmolytes [156].

Yeast extract contains many growth substances—such as vitamins B1, pyridoxine, riboflavin, cytokinins, proteins, and carbohydrates [94,157] and decreases the negative effects of water stress [158]. Yeast plays a significant role in diffusing CO_2_, thus improving photosynthesis, nutrition status, and apple yield and quality [159]. Foliar spraying with yeast extract (*Saccharomyces cerevisiae* 10 g) improved the length and diameter of the roots in olive trees [94].

Garlic possesses sulfur compounds such as trisulfide, aliin, ajoene, allylpropl, allicin, sallylcysteine, and diallyl. In addition to these compounds, garlic possesses different types of amino acids, arginine and glycosides and enzymes such as peroxidases and alliinase [160]. Garlic extract may be used to reduce abiotic and biotic stresses [160]. Foliar spraying with 250 g garlic clove extract, for instance, improved the root growth parameters in olive trees [94].

Fulvic acid is a type of organic acid that plays an important role in increasing plant growth and plant drought resistance [161]. In grape berries, fulvic acid improved fruit quality and calcium absorption [162]. In addition, humic acid is considered a bio-stimulant that improves yield and helps plants resist environmental stresses [163]. Fathy et al. [164] showed that humic acid application in the soil increased the yield and growth parameters of the “Canino” apricot. The benefits of humic acid application, especially in alkaline soils, include enhanced nutrient absorption and increased activity of useful soil microorganisms [165]. The use of humic acid (5mll–1) also decreased pomegranate cracking under drought [123].

Finally, Gibberellins (GAs) are hormones involved in plant growth that regulate different processes of metabolism and gene expression [166]. GA_3_ plays a vital role in mitigating environmental stress [167]. Drought stress during the summer months has been reported to enhance twin fruit development in the following year in sweet cherries [168]. Combined applications of GAs (100 ppm) and nitrogen (2000 ppm) reduced double ovaries during pollination and the twin-fruit percentage under water stress in sweet cherries [169].

## 4. New Molecular Strategies to Combat Drought Stress

The ability of a population to adapt to stressful conditions such as drought and disease depends on the presence of individuals with the gene alleles required to adapt to these conditions [170].

### 4.1. Genome Editing Using CRISPR/Cas9

Plant breeders are now progressively focusing on the latest genome-editing instruments to improve significant agricultural attributes [171]. The emergence of multifold sequence-specific nucleases has simplified accurate gene modification to develop new varieties compatible with climatic changes [172]. Among the existing genome editing technoloies, CRISPR/Cas9 stands out for its pliability, compatibility, broad applicability, and ease of use [173]. The CRISPR/Cas system uses a compound involving a single guide RNA and Cas endonuclease that travel along the DNA strand, causing a double-strand break on the DNA. Afterwards, the breaks are mended by endogenous cell repair [174,175]. Recently, CRISPR/Cas9 technology has effectively been applied to achieve resistance against many environmental stresses, including drought [173]. In tomatoes, slnpr1 mutants were generated to confirm the role of *Pathogenesis Related 1* (*NPR1*) in drought resistance [176]. On the other hand, a decrease in MdNPR1 has been recorded in drought- responsive apple plants [177].

Gene expression changes are usually the first response to stress situations in plants [178]. Among these stress-reactive genes, those that encode transcription factors (TFs) have a significant role in adjusting the plant reaction to the stress situation [179]. *Dehydratation-Responsive Element Binding Factors* (*DREBs*) control the expression of some target genes induced by cold and drought stress [180]. *MsDREB6.2* overexpression leads to a decrease in stomatal density and apertures and an increase in the hydraulic conductivity of roots, which boosts the drought resistance of overexpressing plants [181]. In these plants, the leaves were thicker and, while stem growth was delayed, root growth increased [181]. It is important to note that root increase is characteristic of plants that show drought tolerance [182].

### 4.2. Transcription Factors

Transcription factors (TFs) are the key regulators of gene transcription. Transcription factors also act as proteins that can bind to DNA sequences and regulate transcription [183]. In wild almonds, promoter analysis showed that differentially expressed genes harbor binding sites of *MYB1* and *MYB2* transcription factors, which are involved in the dehydration response through the ABA signaling pathway [184]. In mango, citrus, and *Papaya, basic helix loop-helix* (*bHLH*) TF genes and members of the WD40 protein family have been found to regulate abiotic stress responses, which can provide knowledge for understanding responses under cold, salinity, and water stress [185]. Eukaryotic elongation factor (18 eIF) genes that were expressed under salt stress, osmosis, and low temperatures were identified using transcriptome analysis [186]. Researchers have also investigated the transcription factors regulating drought stress tolerance in papaya, including *CpHSF*, *CpMYB*, *CpNAC*, *CpNFY-A*, *CpERF*, and *CpWRKY* [187]. Huang et al. [188] classified a total of 103 *WRKY* TFs in the pear genome and showed that drought tolerance was improved by *PbWRKY* manipulation. *WRKY* TFs may also therefore play an important role in regulating the water stress response. In addition, *WUSCHEL-related homeobox* (*WOX*) transcription factors are important in plant development processes and evolutionary novelties. Recently, Lv et al. [189] showed that overexpression of *MdWOX13–1* increased the callus weight and enhanced ROS scavenging against drought stress in different Rosacea species including prune (*Prunus domestica*), apple (*Malus domestica*), pear (*Pirus communis*), almond (*Prunus dulics*), peach (*Prunus persica*), mei (*Prunus mume*), and cherry (*Prunus avium*).

### 4.3. RNA Interference

RNA interference (RNAi) is a rapid method to induce gene silencing in a variety of organisms [190]. In peach and almond, a qPCR analysis confirmed the implication of miR156, miR159, miR160, miR167, miR171, miR172, miR398, miR403, miR408, miR842, and miR2275 in the dehydration stress response. Comparison of miRNA expression patterns in the three evaluated genotypes indicated that the peach-almond hybrid ‘GN-15’ showed higher expression levels of specific miRNAs which should be related to the observed drought tolerance [191]. In addition, an RNAi method was used to knock down GH3 genes in apple trees. The Gretchen Hagen3 (GH3) family proteins convert auxin Indole-3-Acetic Acid (IAA) to IAA-amino acids. Under long-term water stress in apple trees, it was found that MdGH3 RNAi plants performed better than wild-type plants and had a higher root-to-stem ratio, higher water use efficiency, and higher photosynthetic capacity [192]. Knocking down six GH3 family genes was also found to enhance drought resistance under water stress conditions in apple trees [193] (Table 1).

## 5. Conclusions

In a context of global climate change, drought conditions are one of the main limiting parameters for plant yield and growth in agriculture around the world and thus impact global food security. To deal with the global drought conditions, different agricultural management strategies have been studied, including mulching, rainwater harvesting, net shading, the use of tolerant rootstocks and early-maturing cultivars, fruit thinning, pruning and regulated DI strategies, as well as the exogenous application of anti-water stress substances such as ascorbic acid, ABA, melatonin, and proline. Such procedures and strategies can maintain orchard growth and productivity, but new technologies can also enhance plant tolerance to abiotic stress. The CRISPR/Cas9 gene editing system, as well as overexpressing drought-responsive genes, contributes to the promise of fruit tree cultivation in arid regions with high yields and quality, increasing the supply of food for humans around the world.

## Figures and Tables

**Figure 1 plants-12-00773-f001:**
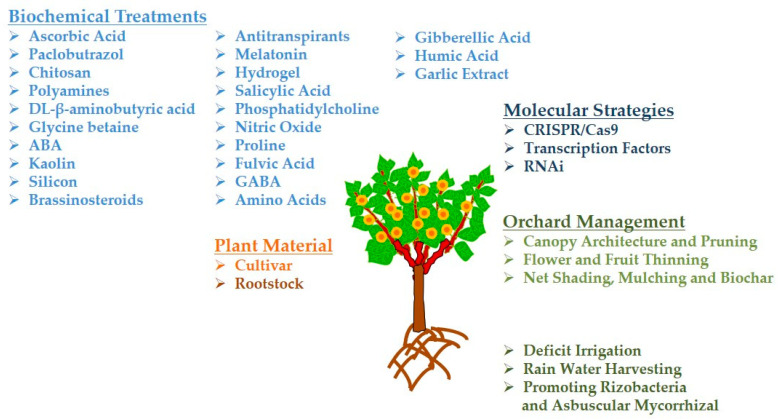
New Horticultural, Biochemical, and Molecular Strategies for Drought Tolerance in Temperate Fruit Crops.

**Table 1 plants-12-00773-t001:** Molecular strategies applied to combat drought in fruit trees.

Species	Target Gene	Trait	Method	References
Almond	*MdWOX13-1*	Drought stress	Transcription factors	[189]
Apple	*MdCIPK6L*	Enhanced tolerance to salt and osmotic/drought stresses	CRISPR	[194]
Apple	*MsDREB6.2*	Increased drought resistance	CRISPR	[181]
Apple	*MdWOX13-1*	Drought stress	Transcription factors	[189]
Apple	*MdGH3* RNAi	Drought stress	RNAi approach	[189,191]
Banana	*MabZIP*	Abiotic stress	Transcription factors	[195]
Cherry	*MdWOX13-1*	Drought stress	Transcription factors	[189]
Mango	*Eukaryotic Translation Initiation Factors* (*eIFs*)	Abiotic stress	Transcription factors	[186]
Mei	*MdWOX13-1*	Drought stress	Transcription factors	[189]
Papaya	*CpHSF*, *CpMYB*, *CpNAC*, *CpNFY-A*, *CpERF* and *CpWRKY*	Drought stress	Transcription factors	[187]
Peach	*MdWOX13-1*	Drought stress	Transcription factors	[189]
Pear	*MdWOX13-1, PbWRKYs*	Drought stress	Transcription factors	[189,196]
Prune	*MdWOX13-1*	Drought stress	Transcription factors	[189]

## Data Availability

Data is contained within the manuscript.

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
