# Peer review of "Orchard Management and Incorporation of Biochemical and Molecular Strategies for Improving Drought Tolerance in Fruit Tree Crops"

_plants, 2023, doi:10.3390/plants12040773_

Round 1

Reviewer 1 Report

The manuscript lists several methods to cope with or ameliorate drought stresses in fruit trees. The authors span from methods used in low-income countries to up-to-date technologies (CRISPR/CAS9), giving an outlook on several methods.

Authors should better describe some water-saving technology like “cocoon” (line 70) and “half-moons” (line 74). In the latter case, check the verb form.

In line 15, the words “especially horticultural crops” are repeated twice.

In line 167, the citations are 37 and 38: while 37 refers to the Morocco Agrosystem, reference 38 refers to “Tree-based intercropping (TBI) systems” in Canada- the reference is incorrect.

In lines 226-236 (RDI), the authors should provide descriptions or examples of the most critical stages.

In line 282 “Kenya” should be substituted with ”in Kenya.”

Reference 115 (Chen, T.H.H.; Murata, N) is present in the References but not in the manuscript text.

Author Response

According with the suggestions of the Reviewer 1 we have revised the manuscript incorporating the proposed revisions indicating these revisions with the control of changes of the WORD document.

We deeply appreciate the efforts of the reviewer in the improvement of the manuscript for a future publication.

Regarding reviewer's 1 comments (R1):

R1: The manuscript lists several methods to cope with or ameliorate drought stresses in fruit trees. The authors span from methods used in low-income countries to up-to-date technologies (CRISPR/CAS9), giving an outlook on several methods.

Authors: We agree and thank the Reviewer 1 for their comments about the revision of this work and for considering this manuscript suitable for future publication. In addition, all the suggestions and revisions of the reviewer have been incorporated indicating these revisions with the “Track Changes” of the WORD document.

 R1: Authors should better describe some water-saving technology like “cocoon” (line 70) and “half-moons” (line 74). In the latter case, check the verb form.

Authors: We agree and thank the Reviewer 1 for their comments. Cocoon and half-moons technologies have been clarified and better explained (lines 75-80).

 R1: In line 15, the words “especially horticultural crops” are repeated twice.

Authors: Revised.

 R1: In line 167, the citations are 37 and 38: while 37 refers to the Morocco Agrosystem, reference 38 refers to “Tree-based intercropping (TBI) systems” in Canada- the reference is incorrect.

Authors: We agree and thank the Reviewer 1 for their comments and revisions. Citation of references 37 and 38 has been revised in the text (lines).

 R1: In lines 226-236 (RDI), the authors should provide descriptions or examples of the most critical stages.

Authors: We agree and thank the Reviewer 1 for their comments. A better description of the most critical RDI stages have been added (lines 240-246).

R1: In line 282 “Kenya” should be substituted with ”in Kenya.”

Authors: Revised.

 R1: Reference 115 (Chen, T.H.H.; Murata, N) is present in the References but not in the manuscript text.

Authors: Reference 115 has been incorporated in line 374.

We deeply appreciate the efforts of the reviewer in the improvement of the manuscript for a future publication.

Yours faithfully,

Dr. Pedro Martínez-Gómez

CEBAS-CSIC, Murcia (Spain)

Reviewer 2 Report

Dear Editor

Thank you very much for your invitation to review this manuscript:

New Strategies for Improving Drought Tolerance in Fruit Tree Crops.

The authors must follow these comments:

This review presents very important information about various strategies for Improving Drought Tolerance in Fruit Tree Crops.

·         The title should be changed to: Management and various strategies for improving drought tolerance in fruit tree crops

·         Lines 144-147, rewrite this sentence.

·         Lines 160-167, One paragraph is better

·         Line 193, subscript

·         Line 201, current reference required

·         Line 237, one paragraph is better

·         Line 260, one paragraph is better

·         Line 275, [75,76]

·         Line 309, Italic

·         Line 316, Italic

·         Line 318, one paragraph is better

·         Line 358, current reference required

·         Line 405, current reference required

·         Line 435, current reference required

·         Line 455, superscript

·         Line 458, one paragraph is better

·         Line 469, subscript

·         Line 479, one paragraph is better

·         Line 486, superscript

·         Line 694, Italic

·         Line 770, Italic

·         Line 794, Italic

·         Lines 815 , Italic

·         See the comments in the Pdf version

Best regards

Author Response

According with the suggestions of the Reviewer 2 we have revised the manuscript incorporating the proposed revisions indicating these revisions with the control of changes of the WORD document.

We deeply appreciate the efforts of the reviewer in the improvement of the manuscript for a future publication.

Regarding reviewer's 2 comments (R2):

R2: This review presents very important information about various strategies for Improving Drought Tolerance in Fruit Tree Crops.

Authors: We agree and thank the Reviewer 2 for their comments about the revision of this work and for considering this manuscript suitable for future publication. In addition, all the suggestions and revisions of the reviewer have been incorporated indicating these revisions with the “Track Changes” of the WORD document.

 R2 The title should be changed to: Management and various strategies for improving drought tolerance in fruit tree crops.

Authors: Title has been revised according to the suggestions of the reviewer as “Management and incorporation of biochemical and molecular strategies for improving drought tolerance in fruit tree crops”.

 R2: Lines 144-147, rewrite this sentence.

Authors: This sentence has been rewritten.

 R2: Lines 160-167, One paragraph is better.

Authors: Merged.

 R2: Line 193, subscript.

Authors: Revised.

 R2: Line 201, current reference required.

Authors: The indicated reference Hafez et al. 2020 Agronomy 10, 630 is a good complementary reference. However unfortunately this work is in Barley and I believe that we cannot include this reference because this review includes only references in fruit tree crops.

 R2: Line 237, one paragraph is better.

Authors: Merged.

 R2: Line 260, one paragraph is better.

Authors: Merged.

 R2: Line 275, [75,76].

Authors: Revised.

 R2: Line 309, Italic.

Authors: Revised.

 R2: Line 316, Italic.

Authors: Revised.

 R2: Line 318, one paragraph is better.

Authors: Merged.

 R2: Line 358, current reference required.

Authors: The indicated reference Alkhatani et al. 2020 Agronomy 8, 1180 is a good complementary reference. However unfortunately this work is in Sweet Peper and I believe that we cannot include this reference because this review includes only references in fruit tree crops.

R2: Line 405, current reference required.

Authors: The indicated reference Abdelaal et al. 2020 Plants 9, 733 is a good complementary reference. However unfortunately this work is in Sweet Peper and I believe that we cannot include this reference because this review includes only references in fruit tree crops.

R2: Line 435, current reference required.

Authors: The indicated reference Abdelaal et al. 2020 Sustainability 12, 1736 is a good complementary reference. However unfortunately this work is in Barley and I believe that we cannot include this reference because this review includes only references in fruit tree crops.

 R2: Line 455, superscript.

Authors: Revised.

 R2: Line 458, one paragraph is better.

Authors: Merged.

 R2: Line 469, subscript.

Authors: Revised.

 R2: Line 479, one paragraph is better.

Authors: Merged.

 R2: Line 486, superscript.

Authors: Revised.

 R2: Line 694, Italic.

Authors: Revised.

 R2: Line 770, Italic.

Authors: Revised.

 R2: Line 794, Italic.

Authors: Revised.

 R2: Line 851, Italic.

Authors: Revised.

 R2: See the comments in the Pdf version.

Authors: We agree and thank the Reviewer 2 for their comments about the revision of this work in the attached PDF. All the suggestions and revisions of the reviewer have been incorporated indicating these revisions with the “Track Changes” of the WORD document. In addition, 5 new refences have been added attending the suggestions of the reviewer.

 We deeply appreciate the efforts of the reviewer in the improvement of the manuscript for a future publication.

Yours faithfully,

Dr. Pedro Martínez-Gómez

CEBAS-CSIC, Murcia (Spain)